# Transgender Males as Potential Donors for Uterus Transplantation: A Survey

**DOI:** 10.3390/jcm11206081

**Published:** 2022-10-14

**Authors:** Marie Carbonnel, Léa Karpel, Ninon Corruble, Sophie Legendri, Lucile Pencole, Bernard Cordier, Catherine Racowsky, Jean-Marc Ayoubi

**Affiliations:** 1Department of Obstetrics and Gynecology, Foch Hospital, 40 Rue Worth, 92150 Suresnes, France; 2Medical School, University of Versailles, Saint-Quentin-en-Yvelines, 55 Avenue de Paris, 78000 Versailles, France; 3Department of Psychiatry, Hospital Foch, 40 Rue Worth, 92150 Suresnes, France; 4Department of Obstetrics, Gynecology and Reproductive Biology, Brigham and Women’s Hospital, Boston, MA 02115, USA

**Keywords:** male, transgender, uterus, transplantation, hysterectomy

## Abstract

Uterus transplantation is a new treatment for patients with absolute uterine infertility that is conducted in order to enable them to carry their own pregnancy. One of the limitations for its development is donor availability. Some transgender males undergo a hysterectomy in the gender-affirming surgery process, and might be interested in donating their uterus for transplantation. In this manuscript, we report the results of a survey designed to determine the attitudes of such individuals regarding donation of their uterus for this purpose. Over 32 years (January 1989–January 2021), 348 biological women underwent hysterectomy at our hospital as part of gender-affirming surgery. The survey was sent to 212 of the 348 prospective participants (for 136, we lacked postal or email addresses). Among the 212 surveys sent, we obtained responses from 94 individuals (44%): 83 (88.3%) stated they would agree to donate, of whom 44 would do so for altruism, 23 for the usefulness of the gesture and 16 out of understanding of the desire to have a child; 63 (75.5%) wanted to know the recipient and 45 (54.2%) wanted to know the result of the donation. According to this survey, a high proportion of transgender males surveyed would be interested in donating their uterus for uterus transplantation.

## 1. Introduction

In Europe, it is estimated that 150,000 women of childbearing age are affected by permanent uterine infertility, either congenital or acquired [1]. Congenital absolute uterine infertility (AUFI), otherwise known as the Mayer–Rokitansky–Küster–Hauser (MRKH) syndrome, involves the absence of the uterus and affects 1 in 4500 females [2]. Acquired AUFI results when a patient undergoes hysterectomy [3]. For these patients, adoption or surrogacy (currently illegal in France and in many countries) represented the only possible ways to become a mother until uterus transplantation became available. 

The feasibility of uterus transplantation was documented in 2014 following the birth in Sweden of a healthy baby after transplantation from a living donor to a recipient with MRKH [4]. Since this date, 80 uterus transplantations have been performed world-wide, and more than 35 healthy births have occurred [5]. This emerging technology has the potential for translation into mainstream clinical practice. However, several limitations to access prevail, one of which is uterus donor availability. 

To date, despite deceased donors being used in some cases, the great majority of uterus donation have been from live donors [6]. The use of live donors enables extensive pre-transplantation evaluation and planning. Living donors have mainly been relatives of the recipients, often their mothers, and so had emotional and genetic relationships with them [7]. However, experience shows that potential related donors have a 75% risk of not fulfilling current inclusion criteria [8]. An alternative to directed living donation is altruistic, non-directed donation. It is currently offered by two teams [9,10]. 

Uterus donation by patients with a normal uterus requiring hysterectomy could also be performed. Furthermore, the surgical operation would not be carried out for the sole purpose of a uterine transplantation. Such is the case for transgender males who decide to have a hysterectomy as part of their gender-affirming surgery. 

The term “transgender” describes persons whose gender identity is incongruent with the phenotypic sex assigned at birth, the latter being generally concordant with the genetically determined sex. Transgender people who follow their inner sense of gender identity often choose to transition to their perceived proper gender identity. In cases of transgender males, the transition is from a woman to a man.

Gender dysphoria is far from being a rare occurrence. There are an estimated 1.4 million transgender individuals in the United States, representing 0.6% of the population [11]. Of the transgender men, 14% have undergone hysterectomy as part of gender-affirming surgery for psychological and medical reasons [12]. In a preliminary Turkish survey of 31 transgender males, 84% stated they would volunteer for uterus donation [13].

In France, transgender patients are treated in specialized units, such as that in our hospital, which has been operational for 32 years. Our center also has a uterus transplantation program, having performed the first uterus transplantation in France [14], resulting in a healthy live birth. We therefore have a special interest in determining the viewpoints of transgender males who undergo hysterectomy in their gender-affirming surgery. In this manuscript, we report the results of a survey designed to determine the attitudes of such individuals regarding donation of their uterus for uterus transplantation.

## 2. Materials and Methods

### 2.1. Institutional Approval

This study was approved by the Foch Hospital Institutional Review Board (IRB 00012437).

### 2.2. Patient Population

During the 32 years through 2021 (January 1989–January 2021), 348 biological women underwent hysterectomy at the Foch Hospital in Suresnes as part of gender-affirming surgery. All patients fulfilled the following criteria for this surgery as required by the Standards of Care (SOC) for the Health of Transsexual, Transgender, and Gender Nonconforming People published by the World Professional Association for Transgender Health (WPATH): experiencing persistent, well-documented gender dysphoria; possessing the capacity to make a fully informed decision and to give consent for treatment; being at the age of legal majority (over 18 years); and having been on at least 12 continuous months of testosterone therapy [15]. All patients underwent the required gender-affirming process involving a multidisciplinary team of gynecologists, psychologists, urologists, plastic surgeons, psychiatrists and endocrinologists. Patients were evaluated for at least two years by psychiatrists and psychologists. At the end of the first year of evaluation, the decision to provide these women with masculinizing hormone therapy was established in a multidisciplinary consultation meeting, and the surgical transformation was also submitted to this same RCP after at least two years. Transformation surgery, called sexual reassignment, involved a mastectomy followed by a total non-conservative hysterectomy if wanted. All the above steps were performed at our hospital.

### 2.3. The Survey

This was a retrospective survey conducted after the surgery was performed to explore whether the respondents would have been willing to donate their uterus if they had been offered the opportunity to do so. The survey was sent by mail or email in January 2021 and was closed in June 2021, i.e., 6 months later. 

The questions asked in this survey are available in Appendix A. The survey consisted of four parts. The first part explored the age and the marital and family situation of the patients. The second part concerned their experience with the hysterectomy from a physical and emotional point of view, including their motivation for removing their uterus (i.e., a sincere wish to remove as opposed to a desire for its removal solely to obtain a change of civil status); and how they experienced the removal (i.e., as an amputation, a relief, a mutilation, a right, a forced sterilization, or other). The third part of the survey concerned their retrospective willingness (or not) to participate in such a transplant project, and the reasons that would have led them to agree or disagree with such a proposal (free-form responses were used in this section). An explanation of protocol variance for hysterectomy in the context of uterus donation was included, emphasizing the increased duration of surgery (10 h), more pre- and post-operative follow-up and more risk of complications, including the possible involvement of ureters, compared with a simple hysterectomy. We also asked them if they wanted to know the recipient (i.e., whether or not their gesture was anonymous) and the outcome of their donation. The final part of the survey concerned their gynecological history before sexual reassignment, in order to evaluate the potential quality of the uterine grafts.

Microsoft Excel was used for data recording. Responses were analyzed using the mean (±standard deviation) for continuous variables and reported as percentages of all responders for categorical variables.

## 3. Results

### 3.1. Response Rate

The survey was sent to 212 of the 348 prospective participants (we were unable to contact 136 because we lacked correct postal or email addresses). Among the 212 surveys sent, we obtained responses from 94 individuals (44%), which represent a total of 27% among transgender males who underwent hysterectomy in our unit (Figure 1).

The mean age of respondents upon receipt of the survey was 43.0 ± 13.0 years versus 30.0 ± 8.0 years at the time of their hysterectomies. The average time from hysterectomy to answering the survey was 13.0 ± 8.0 years.

### 3.2. Marital and Family Stituation

The majority of respondents lived with a partner (*n* = 58; 62%), while 38% were single. Nearly half (*n* = 38; 40.4%) were parents or had a parental function: 19.1% (*n* = 18) were fathers by sperm donation, 14.8% (*n* = 14) were stepfathers, 4.2% (*n* = 4) were adoptive fathers, and 2.1% (*n* = 2) were biological parents prior to their transition. At the time of the survey, one patient was waiting for sperm donation and one patient was in the process of adoption. 

### 3.3. Experience of the Hysterectomy

Of the 94 respondents, 85 (90.4%) had either a very good (*n* = 60) or good (*n* = 25) experience with their hysterectomy. Almost all respondents (94.7%; *n* = 89) sincerely wanted this surgery, including 20 who underwent the surgery solely to obtain a change of civil status. The great majority of respondents (84.5%; *n* = 90) felt that the hysterectomy was painless. The most frequently used term to describe their feeling was relief (80.8%; *n* = 76). Others responded that it was their right (21.2%; *n* = 20), a liberation (14%; *n* = 15) or a rebirth (5.6%; *n* = 6). 

One respondent felt indifferent to the hysterectomy, seven had a bad experience, and one had a very bad experience (three for physical complications and five for psychological reasons). The postoperative complications comprised two hemorrhages (one of which required a repeat operation) a wound infection and poor healing. Four respondents considered the hysterectomy as forced sterilization, one of whom felt that it was an amputation or mutilation. Among these four, three had not sincerely wished to undergo hysterectomy, and one did not stipulate the psychological reason for his bad experience. 

### 3.4. Willingness to Donate Their Uterus for Transplantation 

The distribution of the 94 respondents according to their willingness to donate their uterus for transplantation is shown in Figure 2. The great majority (*n* = 83; 88.3%) indicated that they would have agreed to donate their uterus. Eight (8.8%) would have declined and three (3.2%) were undecided.

The reasons for potential refusal to donate included disagreement with organ donation in general (*n* = 1), the risks of complications (*n* = 7), the lengthening of the management of their transsexualism (*n* = 2), and the impossibility for them to cryopreserve their oocytes before their complete non-conservative hysterectomy (*n* = 1).

The main reasons for the affirmative response were altruism (*n* = 44; 53.0%), the usefulness of the gesture (*n* = 23; 27.7%) and the understanding of the desire of a biological woman to have a child (*n* = 16; 19.3%) (Figure 2).

Examples of comments associated with each of these reasons are as follows: -**Altruism:**

“If it can make someone happy”.

“I agree with organ donation”.

-
**The usefulness of the gesture:**


“If it can be used”.

“It is a healthy organ that I had no use for but that can be used by others”.

Of note, we observed in verbatim responses a feeling that their uterus did not belong to them anyway: (“From the moment you took it away from me, you can do what you want with it”), and the feeling that it was a waste to have thrown away their uterus at the time: “It’s a shame to have thrown it in the garbage”.

-
**The understanding of the desire for a child in a biological woman:**


“We too have suffered from not being able to have a child easily”.

“I can understand a woman’s intense desire to have children”. 

“I think for a woman to be complete, she has to have children. If I was allowed to be a man, then I would have done it to allow a woman to have the right to be a woman.”

-A counter-donation to sperm donation was mentioned by two respondents:

“We were able to be parents thanks to a donation, it is a fair return if I could also help a couple to have a child thanks to my uterus”.

Among those who would have agreed to donate their uterus, 75.5% did not want to know the recipient (*n* = 63). They said: “To each his own” “To respect her confidentiality, her intimacy”. Those who would have liked to know her said: “To know her happiness” “To know the person who has a piece of you”.

Half of the patients (*n* = 45, 54.2%) would have liked to know the result of their donation: One of our patients commented: “To know if I made a woman happy”.

### 3.5. Gynecologic History

The average age of the patients at the time of their surgery was 30.0 ± 8.0 years. Most of the patients (*n* = 76, 80.8%) had never had a gynecological follow-up before the consultations preceding the hysterectomy, and the great majority (*n* = 87, 92.5%) had never had a gynecological pathology nor gynecological surgery (*n* = 91, 96.8) before their hysterectomy. No case of gynecologic cancer was reported. 

## 4. Discussion

This survey shows that a great majority (88%) of the transgender male patients who were surveyed after performing hysterectomy in our center would have agreed to donate their uterus for women with uterine infertility. However, the response rate to our survey was relatively low (44% of surveyed patients and only 27% of operated patients) and so it is unknown whether this reflects the true rate of willingness to donate among all people in this special population. Our low survey response rate was likely caused either by incorrect postal or email addresses due to the long lapse between surgery and survey release (maximum of 32 years); and/or by the difficulty for transgender males in speaking about his uterus, which could remind him of his former status as a woman.

The gender-affirmation process includes emotional, social, medical and surgical steps related to that transition. However, not all transgender people desire medical or surgical intervention. Some transgender men undergo gender-affirming surgery to masculinize their body parts in order to bring their physical appearance into harmony with their gender identity. According to the 2015 United States Transgender Survey (USTS), which surveyed 27,715 respondents, 49% had received gender-affirming hormone therapy (GAHT) and 25% had undergone some form of gender-affirming surgery [16], which has substantially increased over the past 20 years [17]. However, only 14% of transgender men had undergone hysterectomy in the United States, according to a report in 2018 [12].

In France, hysterectomy was mandatory for the approval of birth certificate sex change until 2016 [18]. Since then, the rate of hysterectomy has decreased, as the uterus is an invisible organ that has no influence on the male appearance. However, the presence of the uterus can be unbearable for some transmen for medical or psychological reasons (menstrual cycle issues, or its status as an organ emblematic of feminity). Thus, many medical organizations, including the American College of Obstetricians and Gynecologists (ACOG), regard this surgical procedure as medically necessary for transgender patients undergoing transition. The cost of the procedure is covered in France [19].

The majority of the patients in our survey were in couples (62%). Forty percent of them were raising or had raised children. This is in line with our previous study including 134 transgender men from the same cohort, which found that almost half of the transgender men were living with children [20]. The individuals were mostly fathers thanks to sperm donation, or were stepfathers. In France, transgender men who obtain a legal civil change and have a marital life with a female partner qualify for fully covered sperm donation in order to allow the female partner to become pregnant. Only a few were fathers from adoption. This rate could be due to the low rate of adoptable children and discrimination against transgenderism [21]. Two responders had had children before their transition.

The experience of hysterectomy was good for more than 90% of the responders, and 95% of them wished to proceed with the surgery. Although 21% underwent the surgery solely for approval of their birth certificate sex change, it was a relief for 80% of them. Only a few considered the surgery as forced sterilization or amputation. The fact that hysterectomy was mandatory for civil status change before 2016 may explain these cases. However, clinicians should counsel their transgender patients on the potential for not being able to have a biologically related child, and in turn, discuss the possibility of fertility preservation measures prior to initiating testosterone therapy and/or hysterectomy with or without bilateral oophorectomy [22,23,24,25]. Fertility preservation options include cryopreservation of oocytes and retention of the ovaries and uterus if transmen want to carry a future pregnancy. In practice, however, very few male transgender persons carry a pregnancy or use their own gametes with a surrogate mother once the sex change is made [21]. Of note, in France, surrogacy is forbidden.

A major limitation for the translation of uterus transplantation into clinical practice concerns donor availability. In France, two types of organ donation are possible: either a donation from a brain-dead person, or a donation from a living donor to a recipient, provided that the donor is a relative of the recipient or shares emotional ties. However, few suitable uteri are available from brain-dead donors [26] or from close relatives or long-term friends, because of medical conditions or immune-incompatibility [8]. Nevertheless, as the uterus is a non-vital organ, the sole function of which is to carry a pregnancy, uterus donation may occur after the completion of childbearing. Of note, altruistic, non-directed LD uterus donation has been practiced in trials in the United States and the Czech Republic [10,27].

In cases of hysterectomy for gender-affirming surgery when the uterus is normal, the organ could be re-used for uterus transplantation [28]. This has already been suggested in a Turkish survey showing that 26/31 male transgenders wanted to donate their uterus for uterus transplantation [13]. It could be more acceptable than altruistic donors, for which a surgery is performed only for uterus transplantation. However, these transgender males are often treated with high doses of androgens for prolonged periods before surgery, which may affect the functionality of the uterus, particularly the growth of the endometrium. Restoration of normal functionality of the endometrium is mandatory for uterus transplantation. More evaluation would therefore be needed to evaluate whether uterine functionality could be restored after transplantation. Transgender males are also mainly nulliparous, which is a contraindication of uterine transplantation in many trials [8]. Futhermore, hysterectomy for uterus transplantation is much more extensive and riskier than simple hysterectomy: the risk of major postoperative complications (≥Clavien Dindo III, involving mainly ureters) is estimated at up to 10% in live donor hysterectomy, and the duration of surgery is around 10 h, which relates to the time taken to dissect the uterus with the sufficiently long uterine vessels required to perform the transplantation. This is a serious ethical limitation. The rate of complications and the duration of surgery must decrease before it is ethically acceptable for transgender men to become live uterus donors after hysterectomy for gender-affirming surgery. The risk of complications was the main reason of the refusal to donate in our survey.

Our survey findings showed that the main motivation for donation was altruism and the usefulness of the procedure. However, the understanding of the desire to have a child was also often expressed. The fact that the responders faced difficulties in becoming parents could explain the 19% who revealed special empathy toward women with uterus infertility, answering that this was their motivation for uterus donation. If, in the future, transgender men become candidates for uterus donation, special attention should be given to their acceptance of the definitive infertility caused by hysterectomy, and sperm donation should be extensively discussed. In countries where surrogacy is allowed, oocyte cryopreservation possibilities should be thoroughly explained. Our findings further revealed that patients who would agree to donate their uterus would do so mainly through non-anonymous donation and would like to know the result of their donation. It goes without saying that great caution would be needed to avoid any risk of organ trading or psychological pressure for the donor. 

Questioning transgender males as potential donors for uterus transplantation inevitably raises questions about transgender females as potential candidates for uterus transplantation [29]. There is a significant debate in the uterus transplant community regarding the ethical issues of “taking” from a population that we are not willing to “give” to. Several ethical, legal, anatomical, hormonal, fertility and obstetrical considerations raise the complexity of uterus transplantation in transgender females, and more research and reflection would be required [30].

## 5. Conclusions

This survey concerning 94 transgender males who underwent hysterectomy in their gender affirming surgery process showed that a high proportion of those who answered would agree to donate their uterus for uterus transplantation. Our low rate of answers and loss of follow up patients is nevertheless a serious limitation, and does not allow us to extrapolate the results to our entire population.

## Figures and Tables

**Figure 1 jcm-11-06081-f001:**
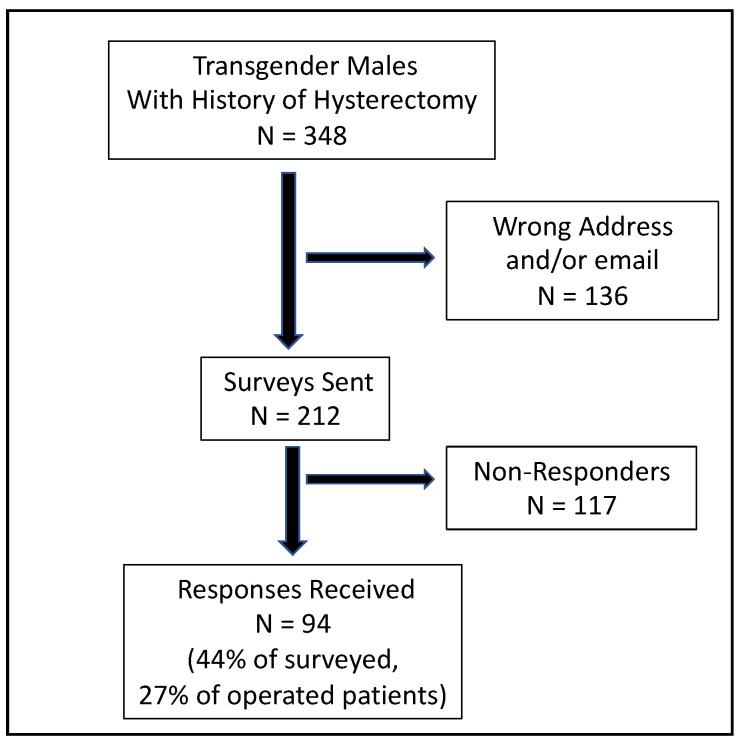
Flowchart of the study.

**Figure 2 jcm-11-06081-f002:**
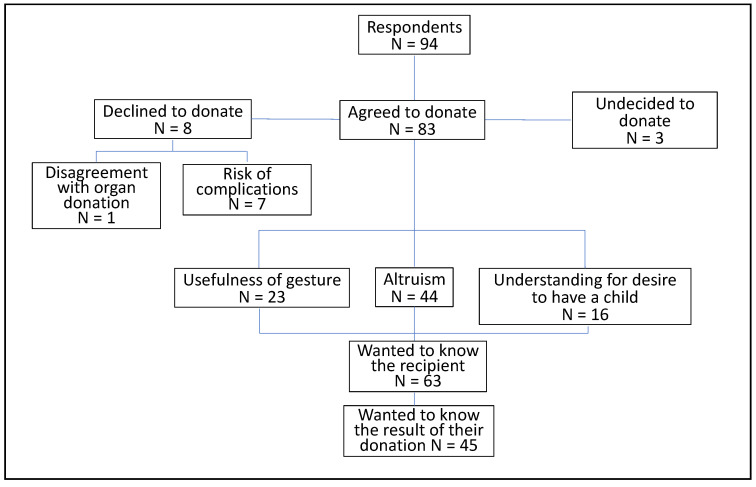
Results of the survey concerning uterus donation.

## Data Availability

Not applicable.

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
