# Peer review of "Transgender Males as Potential Donors for Uterus Transplantation: A Survey"

_jcm, 2022, doi:10.3390/jcm11206081_

Round 1

Reviewer 1 Report

Overall impression:

1.       This is a very interesting topic that deserves a good discussion and review. The authors have a good size center so the research possibilities should be great. Although I congratulate the authors of their attempt, I do have major concerns with the report.

2.       The authors talk about the alternatives of uterus transplantation as being adoption and gestational carrier. The state that “For these patients, adoption or surrogacy, 36 currently illegal in France and in many countries, represented the only possible ways to become a mother until uterus transplantation became available.” Is this really the case that adoption is not legal in France? There is very few cultures and countries that this would be the case. If this statement is true, please provide a reference.

3.       Living donation: The authors state “Living donors have been mainly relatives to recipients, often mothers, and so had emotional or genetic relationships with the recipient, as with the model of kidney transplantation [7]. However, experience shows that potential related donors have a 75% risk of not fulfilling current inclusion criteria [8]. An alternative to directed living donation is altruistic, non-directed donation. Although currently offered by two teams [9,10], it is ethically debatable” This statement is not accurate. With the increasing rate of nondirected donation in the US, Italy, Spain etc most living donors are currently non-directed and not directed. Please rephrase. In addition, please give examples and references to the statement that living donation is unethical.

4.       Study population is an issue. Only 27% (94) out of the surgically treated patients (348) participated in this survey. This is a very skewed sample. It is impossible to draw any conclusions from this population. I would also recommend changes to Figure 1 that suggests that 44% of the 348 patients responded to this questionnaire

5.       I would be interesting to hear the authors explore what has previously been posed as the big hurdles for TM uterus donation. The fact that these individuals are often nulliparous and treated with male hormones prior to surgery is not mentioned at all and might have a bigger impact on their ability to donate than their willingness.

6.       Ethical issues:

a.       There has been a big debate in the uterus transplant community regarding the ethical issues to “take” from a population that we are not wiling to “give” to. Meaning that we do not offer uterus transplantation to the transgender population, but we are considering making them donate. If centers offered transplantations to TF this argument would be easier to meet. The authors do not recognize any of these issues.

b.       A simple hysterectomy performed in an affirmation surgery is miles apart from a uterus donation. The extensiveness and complication rates are not comparable at all. This is also an ethical question that is not touched upon in the manuscript. It is one thing to show willingness to donate your uterus if you think that it is going to be the same surgery. But it is not at all.

Author Response

Response to Reviewer 1

We would like to thank the reviewer for kindly accepting to review our revised manuscript. We also thank the reviewer for the important revisions they suggested. We feel that revisiting and reshaping the manuscript according to the reviewer’s recommendations has greatly improved its readability.

        Point 1 : The authors talk about the alternatives of uterus transplantation as being adoption and gestational carrier. The state that “For these patients, adoption or surrogacy, 36 currently illegal in France and in many countries, represented the only possible ways to become a mother until uterus transplantation became available.” Is this really the case that adoption is not legal in France? There is very few cultures and countries that this would be the case. If this statement is true, please provide a reference.

        Response 1: We apologize for this misunderstanding. It is a problem with punctuation. We rephrased the sentence explaining that only surrogacy is illegal in some countries: Line 36-38: “For these patients, adoption or surrogacy (currently illegal in France and in many countries) represented the only possible ways to become a mother until uterus transplantation became available.

       Point 2: Living donation: The authors state “Living donors have been mainly relatives to recipients, often mothers, and so had emotional or genetic relationships with the recipient, as with the model of kidney transplantation [7]. However, experience shows that potential related donors have a 75% risk of not fulfilling current inclusion criteria [8]. An alternative to directed living donation is altruistic, non-directed donation. Although currently offered by two teams [9,10], it is ethically debatable” This statement is not accurate. With the increasing rate of nondirected donation in the US, Italy, Spain etc most living donors are currently non-directed and not directed. Please rephrase. In addition, please give examples and references to the statement that living donation is unethical.

Response 2: Thank you for your comment.  We rephrase the sentence in order to remove the notion that non directed donation of organ is rare or unethical.

Line 49-57: “Living donors have been mainly relatives to recipients, often mothers, and so had emotional and genetic relationships with the recipient [7]. However, experience shows that potential related donors have a 75% risk of not fulfilling current inclusion criteria [8]. An alternative to directed living donation is altruistic, non-directed donation. It has been currently offered by two teams [9,10].Uterus donation by patients requiring hysterectomy and with a normal uterus could be also performed. Furthermore, the surgical operation would not be carried out for the sole purpose of a uterine transplantation. Such is the case for transgender males who decide to have a hysterectomy as part of their gender-affirming surgery)”.

Point 3: Study population is an issue. Only 27% (94) out of the surgically treated patients (348) participated in this survey. This is a very skewed sample. It is impossible to draw any conclusions from this population. I would also recommend changes to Figure 1 that suggests that 44% of the 348 patients responded to this questionnaire

Response 3: As asked, we modified the figure including the rate of 27% of surgically treated patients. Furthermore, we extensively modified the manuscript in order to emphase the skewed sample:

-in Abstract Line 26-28: “According to this survey, a high proportion of male transgenders surveyed would be interested in donating their uterus for uterus transplantation.”

-In Results, Line 132-134: “Among the 212 surveys sent, we obtained responses from 94 individuals (44%), which represent a total of 27% among transgender males who had hysterectomy in our unit (Figure 1)”.

-In discussion: Line 210-213: “This survey shows that a great majority (88%) of the transgender male patients who were surveyed after performing hysterectomy in our center would have agreed to donate their uterus for women with uterine infertility. However, the response rate to our survey was relatively low ( 44% of surveyed patients and only 27% of operated patients)”.

-And conclusion: line 311-315: “This survey concerning 94 male transgenders who underwent hysterectomy in their gender affirming surgery process, showed that a high proportion of those who answered would agree to donate their uterus for uterus transplantation. Our low rate of answers and lost of follow up patients is nevertheless a serious limitation and does not allow us to extrapolate the results to our entire population.

Point 4: I would be interesting to hear the authors explore what has previously been posed as the big hurdles for TM uterus donation. The fact that these individuals are often nulliparous and treated with male hormones prior to surgery is not mentioned at all and might have a bigger impact on their ability to donate than their willingness.

Response 4: We rephrase the sentences according to your judicious comments: Line 276-282: “However, these transgender males are often treated with high doses of androgens for prolonged periods before surgery, which may affect functionality of the uterus, particularly growth of endometrium. Restauration of  normal functionality of endometrium is a mandatory for uterus transplantation. More evaluation would therefore be needed to evaluate whether uterine functionality could be restored after transplantation. Transgender males are also mainly nulliparous, which is a contraindication of uterus transplantation in many trials [8].”

      Point 5: Ethical issues:There has been a big debate in the uterus transplant community regarding the ethical issues to “take” from a population that we are not wiling to “give” to. Meaning that we do not offer uterus transplantation to the transgender population, but we are considering making them donate. If centers offered transplantations to TF this argument would be easier to meet. The authors do not recognize any of these issues.

Response 5 :  We totally agree with this comment, that’s why we added Line 305-311:  “Questioning male transgenders as potential donors for uterus transplantation inevitably raises questions about females transgender as potential candidates for UTx [29]. There is a big debate in the uterus transplant community regarding the ethical issues to “take” from a population that we are not wiling to “give” to. Several ethical, legal, anatomical, hormonal, fertility, and obstetrical considerations raise the complexity of uterus transplantation for females transgenders and need more research and reflexion would be required [30].” 

 Point 6: A simple hysterectomy performed in an affirmation surgery is miles apart from a uterus donation. The extensiveness and complication rates are not comparable at all. This is also an ethical question that is not touched upon in the manuscript. It is one thing to show willingness to donate your uterus if you think that it is going to be the same surgery. But it is not at all.

Response 6: We completely agree with your comments. We modified the discussion in order to emphase on this point: Line 282-288: “Futhermore, hysterectomy for uterus transplantation is much more extensive and riskier than simple hysterectomy: the risk of major postoperative complications (>Clavien Dindo III, involving mainly ureters) is estimated at up to 10% in live donor hysterectomy and the duration of surgery is around 10 hours, which relates to the time taken to dissect the uterus with sufficiently long uterine vessels required to perform the transplantation. This is a serious ethical limitation so far. the rate of complications and the duration of surgery need to decrease before it is ethically acceptable for transgender men to become live uterus donors from their hysterectomy for gender-affirming surgery.”

Reviewer 2 Report

The authors present a survey of 94 transgender men who underwent hysterectomy at a single institution between 1989-2021. They found that, despite the significantly increased risk of complications for donor hysterectomy relative to simple hysterectomy, 88% would agree to donate. They asked several other useful questions, the responses to which could help guide this area of a transplant surgery. As uterine transplantation and transgender surgery both become more common around the world, these will become increasingly important questions. The manuscript might be improved by considering the following minor points.

3.2: last sentence: “one patient was in the process of donating sperm…” I think this is worded incorrectly if we are talking about transgender men (female to male).

Discussion: paragraph 6; first sentence: The abbreviation “UTx” should be defined the first time it is used. As this is not a commonly used medical abbreviation, it would probably be best to avoid its use altogether.

Discussion: final paragraph: first sentence: “…raises questions about transgender males as potential candidate for UTx.” I think this is worded incorrectly if we are talking about transgender men (female to male).

Author Response

Response to Reviewer 2

The authors present a survey of 94 transgender men who underwent hysterectomy at a single institution between 1989-2021. They found that, despite the significantly increased risk of complications for donor hysterectomy relative to simple hysterectomy, 88% would agree to donate. They asked several other useful questions, the responses to which could help guide this area of a transplant surgery. As uterine transplantation and transgender surgery both become more common around the world, these will become increasingly important questions. The manuscript might be improved by considering the following minor points.

We would like to thank the reviewer for kindly accepting to review our revised manuscript and for the positive comments about it

Point 1 : last sentence: “one patient was in the process of donating sperm…” I think this is worded incorrectly if we are talking about transgender men (female to male).

Response 1: We apologize for this mistake. We modified the sentence: Line 145-147“At the time of the survey, one patient was waiting for sperm donation and one patient was in the process of adoption.

Point 2: Discussion: paragraph 6; first sentence: The abbreviation “UTx” should be defined the first time it is used. As this is not a commonly used medical abbreviation, it would probably be best to avoid its use altogether.

Response 2: We removed this abbreviation and replaced it with uterus transplantation

Point 3: Discussion: final paragraph: first sentence: “…raises questions about transgender males as potential candidate for UTx.” I think this is worded incorrectly if we are talking about transgender men (female to male).

 Response 3: Sorry for this mistake, we talked about female transgender. We rephased it : Line 305-307  “Questioning male transgenders as potential donors for uterus transplantation inevitably raises questions about female transgender as potential candidates for uterus transplantation [29]